# Hypoxia Inhibits Subretinal Inflammation Resolution Thrombospondin-1 Dependently

**DOI:** 10.3390/ijms23020681

**Published:** 2022-01-08

**Authors:** Sara Touhami, Fanny Béguier, Tianxiang Yang, Sébastien Augustin, Christophe Roubeix, Frederic Blond, Jean Baptiste Conart, José Alain Sahel, Bahram Bodaghi, Cécile Delarasse, Xavier Guillonneau, Florian Sennlaub

**Affiliations:** 1Institut de la Vision, Sorbonne Université, INSERM, CNRS, 75012 Paris, France; saratouhami@gmail.com (S.T.); fanny.beguier@gmail.com (F.B.); tianxiang.yang@inserm.fr (T.Y.); sebastien.augustin@inserm.fr (S.A.); chris.roubeix@gmail.com (C.R.); frederic.blond@inserm.fr (F.B.); jbconart@hotmail.com (J.B.C.); j.sahel@gmail.com (J.A.S.); cecile.delarasse@sorbonne-universite.fr (C.D.); xavier.guillonneau@inserm.fr (X.G.); 2Ophthalmology Department, Pitié Salpêtrière University Hospital, Sorbonne Université, AP-HP, 75013 Paris, France; bahram.bodaghi@aphp.fr; 3Department of Ophthalmology, University Hospital, 54000 Nancy, France; 4CHNO des Quinze-Vingts, INSERM-DGOS CIC 1423, 75012 Paris, France

**Keywords:** age-related macular degeneration, hypoxia, macrophages, mononuclear phagocytes, choroidal neovascularization, thrombospondin 1

## Abstract

Hypoxia is potentially one of the essential triggers in the pathogenesis of wet age-related macular degeneration (wetAMD), characterized by choroidal neovascularization (CNV) which is driven by the accumulation of subretinal mononuclear phagocytes (MP) that include monocyte-derived cells. Here we show that systemic hypoxia (10% O_2_) increased subretinal MP infiltration and inhibited inflammation resolution after laser-induced subretinal injury in vivo. Accordingly, hypoxic (2% O_2_) human monocytes (Mo) resisted elimination by RPE cells in co-culture. In Mos from hypoxic mice, Thrombospondin 1 mRNA (Thbs1) was most downregulated compared to normoxic animals and hypoxia repressed Thbs-1 expression in human monocytes in vitro. Hypoxic ambient air inhibited MP clearance during the resolution phase of laser-injury in wildtype animals, but had no effect on the exaggerated subretinal MP infiltration observed in normoxic Thbs1^−/−^-mice. Recombinant Thrombospondin 1 protein (TSP-1) completely reversed the pathogenic effect of hypoxia in Thbs1^−/−^-mice, and accelerated inflammation resolution and inhibited CNV in wildtype mice. Together, our results demonstrate that systemic hypoxia disturbs TSP-1-dependent subretinal immune suppression and promotes pathogenic subretinal inflammation and can be therapeutically countered by local recombinant TSP-1.

## 1. Introduction

Age-related macular degeneration (AMD) is characterized by sizeable deposits of lipoproteinaceous debris called soft drusen (early AMD), choroidal neovascularisation (wet AMD, late form), or by an extending lesion of the retinal pigment epithelium (RPE) and photoreceptors (geographic atrophy, GA, late form) [1].

A common feature of early and both advanced forms of AMD is their association with chronic accumulation of mononuclear phagocytes (MPs) in the subretinal space, which is physiologically immunosuppressive and devoid of MPs, at least in part due to the presence of immune-suppressive retinal pigment epithelium cells (RPE) [2]. The MP infiltrate is composed of displaced resident macrophages (Mφ), such as microglial cells (MCs) and choroidal Mφs, and monocyte (Mo)-derived inflammatory Mφs (iMφs) [3]. Functional studies in animal models show that the subretinal accumulation of iMφs play a critical role in neovascularization and photoreceptor degeneration that characterize late AMD [2]. However, the reasons for the establishment of non-resolving pathogenic inflammation in AMD is not clear, but may be found in the downstream consequences of AMD-risk factors.

AMD is a common, complex disease that results from the interplay of age, environmental risk factors and genetic variants [2]. We recently demonstrated that the two main genetic AMD-risk variants, CFH Y402H and a haplotype of 10q26, curb TSP-1 activation of CD47 that is necessary for homeostatic MP elimination and the resolution of acute inflammation in the subretinal space [4,5]. Accordingly, TSP-1 deletion leads to increased subretinal inflammation accompanied by increased CNV in laser-injured mice [5,6,7].

Hypoxia, among other contributing mechanisms such as oxidative stress and dysfunctional autophagy, has long been supposed to be an important trigger of AMD [8]. Late AMD, in particular wet AMD is preceded by choriocapillary dropout, reduced ocular blood flow, and drusen deposits that impede with the oxygen supply from the choroid to RPE and photoreceptors [9]. From a systemic point of view, AMD is associated with hypertension, atherosclerosis, cardiovascular disease [10], and emphysema [11] that are all associated with systemic hypoxia [12]. Indeed, hypoxia can activate MPs and induce inflammatory cytokines such as CCL2, TNFα, IL-1β, and IL-6 [13], all shown to be implicated in subretinal inflammation [3,14,15,16,17,18]. Pre-activation of circulating Mos promoting subretinal chronic inflammation could therefore be one of the mechanisms that link systemic hypoxia to AMD.

Here we show that TSP-1 is the most downregulated transcript in monocytes of mice exposed to systemic hypoxia, which strongly reduces the elimination of subretinal MPs. Using TSP-1-deficient mice and recombinant TSP-1, we demonstrate that the hypoxia induced inhibition of MP elimination is dependent on downregulation of TSP-1. Importantly, we demonstrate that intravitreal injections of recombinant TSP-1 completely reversed the pathogenic effect of hypoxemia, accelerated inflammation resolution, and inhibited CNV in vivo.

## 2. Material and Methods

### 2.1. Laser-Injury Model, Intravitreal Injections, and Hypoxia

Male C57BL6/J animals (Janvier-labs, Le Genest-Saint-Isle, France), aged 10 to 12 weeks, were used in this study. Wild type and Thbs1^−/−^ mice were obtained from the Jackson laboratories. All mice were either negative or backcrossed to eliminate the *Pde6b^rd1^*, *Gnat2^cpfl3^*, and *Crb1^rd8^* mutations. Animals were housed in the animal facility under specific pathogen-free condition, in a 12/12 h light/dark (100–500 lux) cycle with water and normal diet food available ad libitum. Laser-coagulations were performed with a Vitra Laser (Quantel Medical, Coumon -d’Auvergne, France) mounted on a surgical microscope as previously described [19]. Briefly, four equidistant impacts (532 nm, 450 mW, 50 ms, and 250 µm) per eye were applied in the mid-periphery. In certain experiments the eyes were intravitreally injected 4 and 7 days after photocoagulation using glass capillaries (Eppendorf, Hamburg, Germany) and a microinjector, with either 2 µL of PBS, or recombinant human TSP-1 (R&D Systems, Minneapolis, MN, USA, 10µg/mL = 80 nM). The mice were sacrificed at the indicated time points and immune-stained RPE/Choroidal flatmounts were analyzed. Ambient hypoxia 10% O_2_/90%N_2_ was administered using an Oxycycler for the indicated times. Control mice breathing normoxic air were housed in the same room under identical conditions. For qPCR analyses, mice were exposed to hypoxia for 40 h then euthanized and cell sorting was performed immediately after the end of the hypoxic challenge as previously described [4]. All experimental procedures were approved by the Ministere de l’éducation nationale, de l’enseignement supérieur et de la recherche (APAFIS#2636-2015110914346299v2).

### 2.2. Monocyte RPE Co-Cultures

Monocyte/RPE co-cultures were performed as previously described [16,17]. Briefly, primary porcine RPE cells were seeded at a density of 75 000 cells/well in DMEM-FCS20%-PS1% and cultured for 4 days before use. We previously characterized these primary RPE cell cultures, showing that they form tight junctions and phagocytose photoreceptor outer segments, stain positive for ZO1, and express retinol dehydrogenase 5, transthyretin, transferrin [16,17]. Monocytes were prepared from human blood from healthy volunteers after written informed consent (approved by the Direction Générale pour la Recherche et l’Innovation of the Ministère de l’Enseignement et de la Recherche (Dossier n°14.007) and by the Commission Nationale de l’Informatique et des Libertés (N/Ref.: IFP/MKE/AR144088)). Briefly, CD14^+^ peripheral blood Mos were isolated by negative selection using the EasySep Human Monocyte Enrichment Cocktail (StemCell Technologies, Saint Egrève, France) as previously described [5]. In vitro hypoxia conditions were created using a hypoxia incubator chamber (Stemcell technologies) filled with 2% O_2_, 5%CO_2_, and 93% N_2_, and de-pressurized after 1 h of culture (hypoxic conditions) and compared to standard normoxic cell culture conditions (5%CO_2_, 19.9% O_2_, 75.1%N_2_). RPE cells were serum-starved for 24 h prior to co-culture with hMos (100 000 cells/well) under normoxic or hypoxic conditions for 24 h. In some experiments each cell type was pre-incubated for 24 h in normoxic or hypoxic conditions for 24 h followed by 24 h normoxic co-culture. Ultra-low adherence surface 96-well culture plates (Corning, Amsterdam, Netherlands) were used throughout to permit Mo transfer after 24 h of pre-incubation by gently pipetting them off the low adherence culture plates. After culture, the plates were fixed in 4% PFA and stained with goat anti-human OTX2 (R&D, 1/500) (specific of RPE cells) and rabbit anti-human hematopoietic transcription factor PU-1, 1/200 (specific of mononuclear phagocytes) and nuclei were counterstained with Hoechst (1/1000, Sigma-Aldrich, Saint-Quentin-Fallavier, France) as previously described [16,17]. Twenty-five fields per well were analyzed and recorded using the Arrayscan software (HCS iDev Cell Analysis Software, Thermo Fisher Scientific, Les Ulis, France).

### 2.3. Gene Expression Analysis

For whole transcriptome analysis, Ly6C^high^ bone-marrow monocytes sorted from normoxic and 40 h hypoxic mice were prepared. After cell lysis, RNA was extracted using the Qiagen RNA Mini Kit with RNase (ribonuclease)–free DNase (deoxyribonuclease) I digestion. RNA quality and quantity were evaluated using BioAnalyzer 2100 with the RNA 6000 pico Kit (Agilent Technologies, Leuven, Belgique). RNA sequencing libraries were constructed from 200 ng of total RNA using a modified TruSeq RNA Sample preparation kit protocol. Pass-filtered reads (using Trimmomatic) were mapped using HiSAT2 and aligned to human reference genome GRCh38.95 [20]. The count table of the gene features was obtained using HTSeq. Normalization and differential expression analysis values were computed with DESeq2 [21]. TPM were determined using Libinorm using htseq mode [22]. Protein coding mRNAs with greater than 100 TPM in the normoxic group and a false discovery rate < 0.05 were selected. For reverse transcription and real-time quantitative polymerase chain reaction (RT-qPCR) total RNA from human and mouse Mos, mouse MCs, and eye-cup were extracted and PCR was performed using StepOne Plus real-time PCR system (Applied Biosystems) as previously described [4]. Results were normalized using house-keeping gene RPS26. PCRs were performed in 45 cycles of 15 s at 95 °C, 45 s at 60 °C. Primers for RT-PCR were purchased from IDT technology (primer sequences at request). 

### 2.4. Immunohistochemistry, CNV and MPs Quantification 

RPE and retinal flatmounts were stained and quantified as previously described [3] using polyclonal rabbit anti- IBA-1 (Wako, Neuss, Germany) and goat anti-mouse ColIV (Biorad, Mitry-Mori, France; 1/100) and appropriate secondary antibodies and counterstained with Hoechst if indicated. Preparations were observed with a fluorescence microscope (DM5500, Leica, Nanterre, France). 

### 2.5. Statistical Analyses

Graph Pad Prism 7 (GraphPad Software) was used for data analysis and graphic representation. All values are reported as mean  ±  SEM. Statistical analyses were performed by Mann–Whitney test for comparison of mean values. The *n* and *p*-values are indicated in the figure legends.

## 3. Results

### 3.1. Hypoxia Increases the Infiltration of Subretinal MPs after Laser-Injury

The subretinal space can be visualized by flat-mount preparations and is physiologically avascular and devoid of IBA1^+^MPs. Laser-injury induces the infiltration of subretinal IBA1^+^MPs, with a maximal recruitment three to four days after the injury, followed by an inflammation resolution phase characterized by dwindling MP numbers and choroidal neovascularization (CNV) formation [14]. 50–70% of the lesional MPs are derived from circulating Mos and the remainder from resident macrophages (microglial cells and choroidal macrophages) [5,23]. Depletion of circulating Mo [5,23,24] and inhibition of Mo-recruitment [25,26,27,28,29] very significantly inhibits CNV formation. Importantly, the infiltrating MPs are observed within the injured tissue in close contact with the forming CNV, but also in the subretinal space, adjacent to physiologically immunosuppressive RPE cells, that are potent inducers of MP death [4,14,15,16].

To evaluate the influence of hypoxia on subretinal neuro-inflammation we exposed laser-injured mice to 10% or 20.9% of ambient oxygen. As young, otherwise healthy, mice quickly adapt to hypoxia, increasing their hematocrit [30] we either exposed the mice during the early phase (d0–4) or during the inflammation resolution phase, when MP elimination exceeds the recruitment rate (d4–10; experimental design Figure 1A). Quantification on RPE/choroidal flatmounts of subretinal IBA-1^+^ (green) MPs on Collagen-4^+^ (Coll4^+^, red) CNV and on the surrounding RPE (0–500 μm from the lesion) revealed that hypoxia did not significantly alter the density of subretinal MPs in the RPE-denuded lesions, directly adjacent to the endothelial cells of the CNV. However, it significantly increased the accumulation of IBA1^+^MPs that accumulate on the immunosuppressive RPE at d4 (190% increase) and induced a continued accumulation of MPs from d4 to d10, when the MP accumulation started to resolve in normoxic mice (460% increase at day 10; Figure 1B–D). Additionally, the size of Coll4^+^CNV was significantly increased at d10 (Figure 1E).

To test whether hypoxia altered RPE-induced Mos elimination in vitro, we incubated equal numbers of peripheral blood Mos with primary RPE cells for 24 h under normoxic (20.9% O_2_) or hypoxic (2% O_2_) conditions and stained the co-culture with an anti-PU1- and anti OTX2-antibodies that allows the differentiation of PU1^+^OTX2^neg^ Mo and OTX2^+^PU1^neg^RPE cells in this cell culture system. Automated quantification of PU1^+^Mos and OTX2^+^RPE cells revealed that hypoxic conditions significantly increased the number of surviving Mos after 24 h of co-culture, but had no effect on the number of RPE cells (Figure 1E). Interestingly, 24 h hypoxic pre-incubation of Mo (Figure 1F), but not RPE cells (Figure 1G), followed by 24 h normoxic co-culture, similarly increased the number of remaining PU1^+^Mos. 

Taken together, our experiments show that systemic hypoxia, induced by reduced ambient air oxygen, is sufficient to significantly increase subretinal MP accumulation and CNV after laser-injury. Interestingly, these differences were observed in the RPE-adjacent MP population surrounding the injury only and were more pronounced when the hypoxia was administered during the inflammation resolution phase. These results suggested that hypoxia interferes with the RPE-induced MP elimination. Our in vitro results show that hypoxia did not diminish immunosuppressive signals from the RPE but significantly increased Mos resistance to RPE-induced elimination.

### 3.2. Hypoxia Decreases Thbs1-Expression in Mos

To identify potential hypoxia-induced downstream mediators of increased Mos resistance to RPE-induced elimination we sequenced the transcriptome of FACS sorted CD45^+^CD11B^+^Ly6G^neg^Ly6C^high^ Mos from bone marrows of 3-month-old mice that were raised in room-air or had been exposed to 10% ambient O_2_ for 40 h (Figure 2A). A 40 h exposure time was chosen for this experiment as normal mice quickly adapt to hypoxic conditions and to detect differences that occur prior to the inflammatory changes observed in Figure 1. In the transcripts that are robustly expressed in normoxic Mo (>100 transcripts per million), our analysis identified 18 transcripts that were more than two-fold overexpressed and 19 transcripts that were more than two-fold underexpressed in Mos from hypoxic mice compared to Mos from room air raised mice.

Interestingly, the most downregulated gene in Ly6^high^Mos from hypoxic animals was thrombospondin 1 (Thbs-1, TSP-1) that we previously showed is necessary for homeostatic MP elimination and the resolution of acute inflammation in the subretinal space [4,5]. Quantification by RT-PCR of *Thbs1*-mRNA on extracts from retina/RPE/choroid, magnet-sorted retinal MCs, and magnet-sorted bone-marrow Mos from room-air control animals and mice exposed for 40 h to 10% O_2_ hypoxia revealed that the hypoxic conditions did not alter the expression in the chorioretinal tissue or resident MCs, but confirmed the significant downregulation of *Thbs1*- transcription in Ly6^high^Mos (Figure 2B). Additionally, the strong induction of THBS1 in freshly (3 h and 12 h) cultured human Mos (hMos) was significantly blunted when the early differentiating hMos were cultured in 2% O_2_ (Figure 2C).

These results suggest that Mo exposed to hypoxia in vitro and from hypoxic animals (Figure 1) might have a reduced capacity to activate their CD47 receptor, due to decreased TSP-1-expression, necessary for their RPE-induced elimination from the subretinal space.

### 3.3. TSP-1 Is Necessary for Hypoxia-Induced Inhibition of Inflammation Resolution after Laser-Injury

To evaluate whether TSP-1 is required for hypoxia to impair inflammation resolution we submitted wildtype-, and Thbs1^−/−^- mice to laser injury and exposed the mice to ambient hypoxia during inflammation resolution (d4–d10) or kept them in room air. In room air raised mice, the quantification of IBA-1^+^ (green) MPs (counted on the RPE at a distance of 0–500μm to Coll4^+^CNV, red staining) on RPE/choroidal flatmounts at d10, revealed a significantly greater number of subretinal MPs in Thbs1^−/−^-mice (Figure 3A) in accordance with their role in subretinal MP elimination we previously described in laser-injury [5], and in subretinal adoptive transfer experiments, after a light challenge and with age [4]. As shown in Figure 1, hypoxia during inflammation resolution significantly increased the number of subretinal MPs in wildtype-mice at d10, to numbers observed in Thbs1^−/−^-mice kept in room-air. However, hypoxia failed to further increase the elevated levels in Thbs1^−/−^-mice, showing the necessity of the presence of TSP-1 for systemic hypoxia to impede inflammation resolution in our experimental conditions. In terms of Coll4^+^CNV formation in room-air exposed animals, we observed increased CNV formation in Thbs1^−/−^-mice compared to wildtype mice as previously described [5,7], but again failed to alter the extent of CNV in Thbs1^−/−^-mice (Figure 3B).

Next, we next injected wildtype mice intra-vitreally four days after laser-injury with PBS or recombinant TSP-1 (rTSP-1), and exposed them to hypoxia until sacrifice at d10. Quantification of IBA-1^+^ (green) MPs surrounding the lesion on RPE/choroidal flatmounts at d10, revealed that the local injection of rTSP-1 completely reversed the effect of hypoxia during the inflammation resolution phase (Figure 3C). Concomitantly, the intravitreal administration of rTSP-1 significantly reduced the exaggerated CNV in this condition (Figure 3D).

These experiments strongly suggest that the promotion of pathogenic subretinal inflammation by systemic hypoxia is at least in part mediated by the downregulation of TSP-1, which inhibits subretinal MP elimination. They also demonstrate that the pro-inflammatory effect of systemic hypoxia can be reversed pharmacologically by local drug administration.

## 4. Discussion

There are many reasons to suspect that local hypoxia plays a role in the pathogenesis of AMD: drusen and pseudo-drusen deposits increase the distance of the choriocapillaries to the photoreceptors, the choriocapillary blood flow is reduced in intermediate and late AMD, and CNV is preceded by choriocapillary dropout [9]. Intriguingly, there might also be a role for systemic hypoxia as AMD is associated with hypertension, atherosclerosis and cardiovascular disease [10] that are all associated with systemic hypoxia [12]. Hypoxic pre-activation of circulating Mos [13] could thereby promote subretinal chronic inflammation and AMD. We here show that hypoxic ambient air (10% O_2_) was sufficient to increase the infiltration induced by laser injury in vivo. Experimental mice were exposed to not more than six days of hypoxia in either the “recruitment”- or “resolution”-phase, as our preliminary data showed that longer-term exposition to 10% ambient O_2,_ air had only marginal effects on subretinal inflammation and CNV likely because these young, healthy mice adapt to experimental hypoxia by erythrocytosis [30] similar to altitude adaptation. Interestingly, hypoxia increased the subretinal MP population that accumulates in the surrounding of the injury, in direct contact with the immune-suppressive RPE, but not in the number of MPs directly adjacent to the endothelial cells of the CNV. Hypoxic exposure following the peak infiltration at d4 when inflammation resolution occurs in room-air, completely prevented the decrease of MP numbers and let to a further increase of the lesion-surrounding MPs at d10 that was accompanied by more sizeable CNV formation (Figure 1). These results suggested that RPE-induced MP elimination might be affected during hypoxia. We previously showed that MPs are quickly eliminated by the RPE in adoptive transfer experiments of Mo and MCs to the subretinal space and in Mo-RPE cocultures in vitro, and that their activation, due to Cx3cr1-deficiency, exogenous APOE, HTRA1, or LPS, significantly increased their resistance to RPE-induced death [4,5,15,16]. Similarly, we here show that hypoxia significantly reduced RPE-induced human Mo elimination in co-culture in vitro and that this effect was due to an effect of hypoxia on Mos rather than the RPE (Figure 1). 

Hypoxia can activate macrophages via the transcription factors hypoxia-inducible factor 1α and 2α (HIF-1α, HIF-2α), NF-κB and AP-1 that is formed by c-Jun and c-Fos [31,32,33]. Intriguingly, the regimen also led to a significant decrease in *Thbs*1- transcripts in Mo (Figure 2). These alterations might be due to c-jun, which is increased by hypoxia [34] and represses *Thbs1*-tanscription [35].

We recently demonstrated that TSP-1 activation of CD47 sensitizes infiltrating subretinal MPs to RPE induced death and elimination [4,5]. To test whether the hypoxia induced deregulation of TSP-1 could be responsible for the inhibition of the elimination of MPs in contact with the RPE we first compared post-injury inflammation resolution in normoxic and hypoxic Thbs1^−/−^-mice to wildtype animals. At d10 room air raised Thbs1^−/−^-mice, revealed a significantly greater number of subretinal MPs in accordance with its role in subretinal MP elimination others and we previously described [4,5,6] (Figure 3). The increased inflammation in Thbs1^−/−^-mice was accompanied by increased CNV corroborating previous results [5,6,7]. Contrary to wildtype mice, hypoxic conditions failed to alter subretinal MP infiltration or CNV in Thbs1^−/−^- mice, showing that it is strictly necessary for the pathogenic effect of systemic hypoxia. Indeed, when we locally replaced TSP-1 in laser-injured mice, we were able to prevent the exaggerated neuro-inflammation and neovascularization induced by the systemic hypoxia (Figure 3). 

In our experiments we exposed the mice to relatively short periods (4 to 6 days) of constant 10% O_2_ hypoxia. This experimental design allowed us to apply hypoxia separately to the “recruitment”- or “resolution”-phase. Ten percent O_2_ constant hypoxia corresponds to the O_2_ partial pressure on Kilimanjaro (5895 m, 5 January 2021: https://hypoxico.com/altitude-to-oxygen-chart/) that tourists frequently climb without oxygen supplementation, and can therefore be considered severe, but not extreme. Interestingly, a recent analysis of a cohort of 67 786 sleep apnea syndrome (SAS) patients, has shown a strong association of chronic systemic intermittent hypoxemia that characterizes this condition, with wet AMD [36]. Although chronic intermittent hypoxia differs significantly from the chronic hypoxia in emphysema or our experimental conditions, it is interesting to note that SAS patients also display reduced serum TSP-1 levels [37] and we observed decreased TSP-1 transcription under hypoxic conditions in our experiments. Future studies using intermittent hypoxia and chronic hypoxia models are needed to determine similarities and differences between the two and also help to understand whether other AMD-risk factors such as age or genetic predispositions impede the adaptation to chronic hypoxia that we observed in young mice and which might prevent AMD in younger patients with chronic hypoxia.

Taken together, our study shows that systemic hypoxia leads to the activation of monocytes, the deregulation of TSP-1 expression and ensuing increased subretinal inflammation and production of pathogenic cytokines. Our study provides rationale for the implication of hypoxia and in particular systemic hypoxia in neuro-inflammation in AMD, and opens avenues toward therapies inhibiting pathogenic chronic inflammation in late AMD. Most importantly, our study shows that local TSP-1 injection can efficiently counter the effect of systemic hypoxia and inhibit pathogenic chronic inflammation in late AMD. A similar approach might also be beneficial in other inflammatory degenerative diseases that are associated with chronic systemic hypoxia, such as atherosclerosis [38] or Alzheimer’s disease [39].

## Figures and Tables

**Figure 1 ijms-23-00681-f001:**
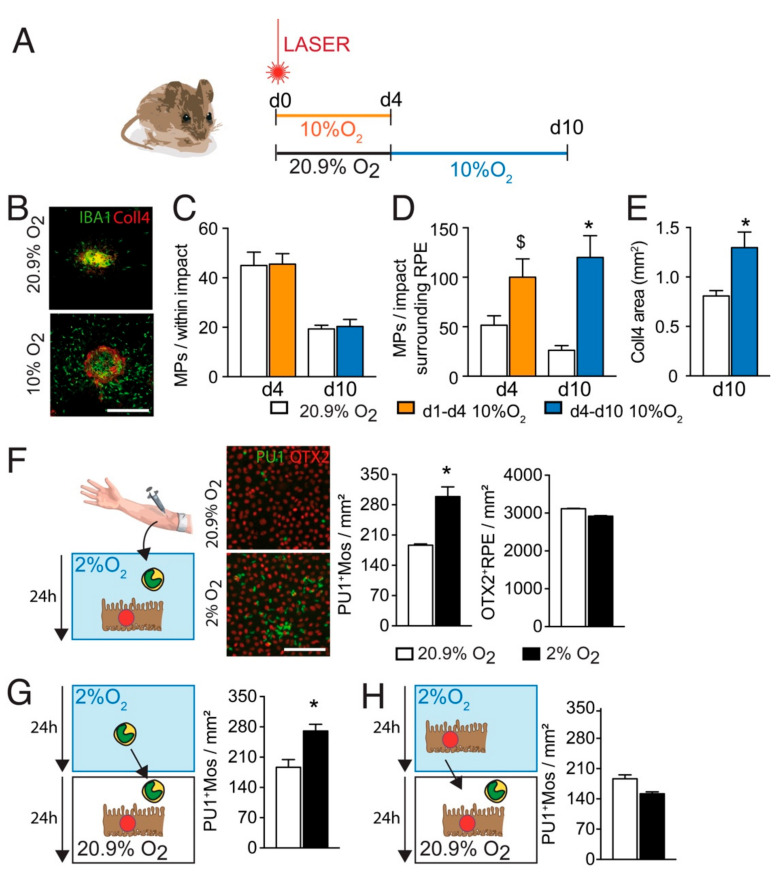
Hypoxia increases Mo resistance to RPE-induced elimination. (**A**) schematic representation of the experiments: mice were laser injured and either exposed to 10% O_2_ hypoxia from d0–d4 and evaluated for the presence of subretinal IBA1^+^MPs at d4 or exposed to 10% O_2_ hypoxia from d4–d10 and evaluated for MPs and the extent of Collagen 4^+^CNV at d10. Mice raised in normal 20.9% ambient O_2_ served as controls. (**B**) Representative images of Collagen 4 (Coll4; red) and IBA-1 (green) immuno- stained RPE/choroid flatmounts of 10 days post laser-injured normoxic and hypoxic (d4–d10) mice. (**C**,**D**) Quantification of the density of subretinal IBA-1+ MPs per impact: (**B**) directly within the RPE-denuded lesion and (**C**) counted at a distance of 0–500μm to Coll4^+^CNV on the apical side of the RPE surrounding the lesion at d4 and d10 of normoxic and hypoxia-exposed 2-month-old mice (room air white columns, 10% O_2_ hypoxia d0-d4 orange columns; 10% O_2_ hypoxia d4–d10 blue columns) (*n* = 9–10 eyes; ^$^
*p* = 0.0503; * *p* < 0.0001 Mann–Whitney versus their normoxic controls). (**E**) Quantification of the Coll4^+^CNV surface at d10 of normoxia and hypoxia-exposed (d4–d10) laser-injured mice (*n* = 9–10 eyes; * *p* = 0.0060 Mann–Whitney versus normoxic control). (**F**) Representative pictures of PU-1 OTX-2 co-stained co-cultures of PU1^+^human monocytes (green) and OTX-2^+^RPE cells (red) under normoxic-(20% O_2_; white columns) and hypoxic-(2% O_2_; black columns) conditions and their automated quantifications after 24 h co-culture (*n* = 5 wells; * *p* = 0.0079 Mann–Whitney versus the normoxic condition), (**G**) quantifications of PU1^+^human monocytes after 24 h of hypoxic (black column) or normoxic (white column) pre-incubation of Mo followed by 24 h of Mo/RPE coculture (*n* = 5 wells; * *p* = 0.0079 Mann–Whitney versus the normoxic condition) (**H**) automated quantifications of PU1^+^human monocytes after 24 h of hypoxic (black column) or normoxic (white column) pre-incubation of RPE followed by 24 h of Mo/RPE coculture. The in vivo results presented in (**B**–**E**) summarize two independently carried out experiments, the in vitro experiments (**F**–**H**) were repeated a minimum of five times and gave similar results. All values are reported as mean  ±  SEM. IBA-1: ionized calcium adapter molecule 1; Coll4: Collagen 4; PU1: hematopoietic transcription factor; OTX-2: Orthodenticle Homeobox 2; scale bar A = 400 μm, E = 500 μm.

**Figure 2 ijms-23-00681-f002:**
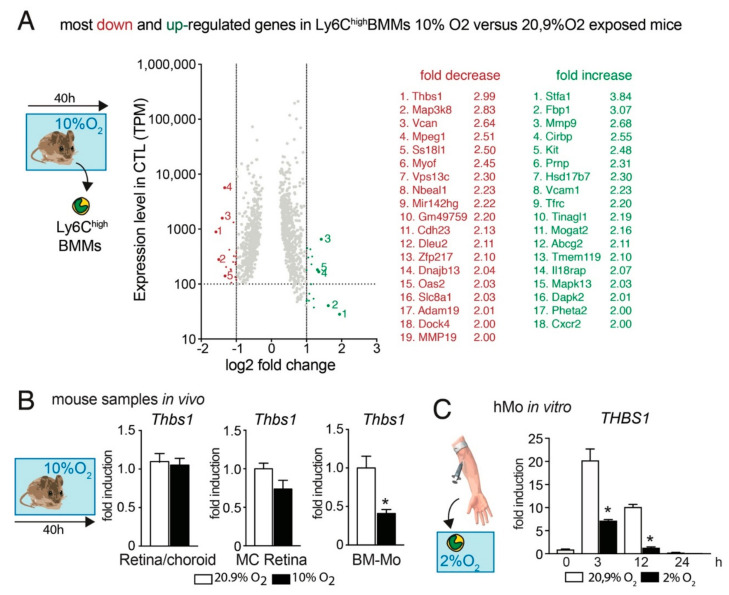
Hypoxia decreases Thbs1- expression in Mos. (**A**) Scatter dot blot of the protein coding mRNAs of FACS sorted CD45^+^CD11B^+^Ly6G^neg^Ly6C^high^ Mos from bone marrows of five 3-month-old mice that were raised in room-air or five age-matched mice that had been exposed to 10% ambient O_2_ for 40 h. Only transcripts with a TPM greater than 100 in the normoxia group and a false discovery rate (FDR) smaller than 0.05 are depicted. The transcripts are plotted according to their expression levels (*y*-axis) and the log2-fold induction by hypoxia (x axis). The identified 18 transcripts that were more than two-fold overexpressed and the 19 transcripts that were more than two-fold under-expressed in Mos from hypoxic mice compared to Mos from room air raised mice are indicated. (**B**) Quantitative RT-PCR of Retina/RPE/Choroid, magnet sorted retinal microglial cells (MC), and magnet sorted bone marrow monocytes (BM-Mo) from room-air-(white columns) and 40 h hypoxia- (10% O_2_; black columns) exposed mice. (*n* = 5 mice/group * *p* = 0.0032; Mann–Whitney versus the normoxic control). (**C**) Quantitative RT-PCR of human Mo exposed to normoxic culture conditions (white columns) or the indicated hours of 2% O_2_ (black columns) (*n* = 5 wells/group, representative of two independent experiments, * *p* = 0.0079 (3 h) and 0.0159 (12 h); Mann–Whitney versus the normoxic control). All values are reported as mean  ±  SEM. Thbs1: Thrombospondin 1.

**Figure 3 ijms-23-00681-f003:**
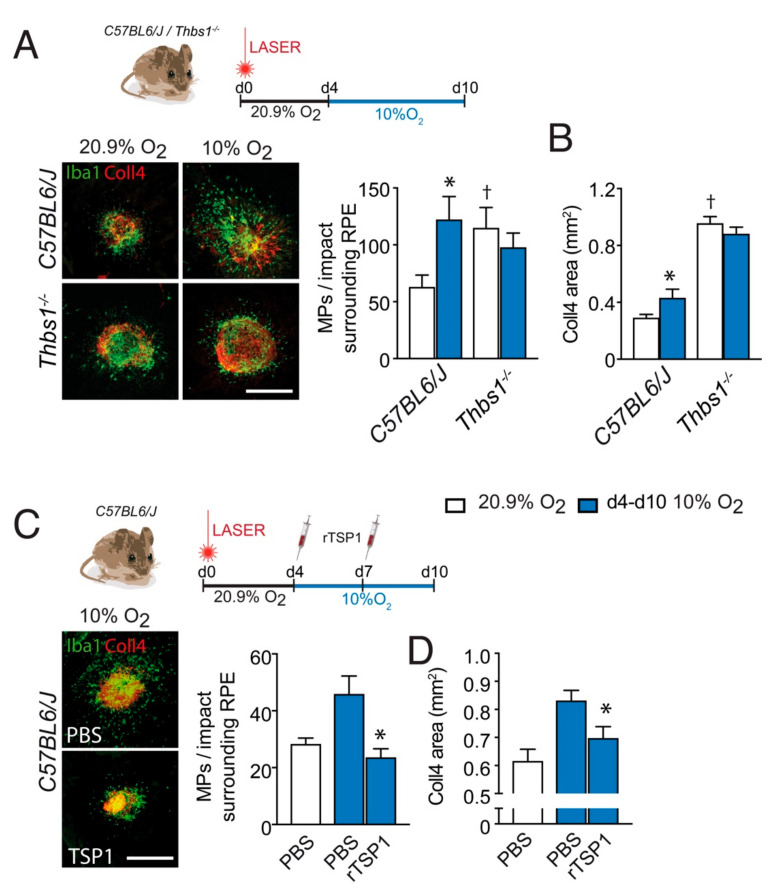
TSP-1 is necessary for hypoxia-induced inhibition of inflammation resolution after laser-injury. (**A**) Representative images of Collagen 4 (Coll4; red) and IBA-1 (green) immuno-stained RPE/choroid flatmounts of 10 day post laser-injured 2-month-old mice of the indicated strains, raised in 20.9% ambient O_2_ (normoxia, white columns) or exposed to 10% O_2_ hypoxia (d4–d10, blue columns) and quantification of subretinal IBA-1^+^MPs per impact on the RPE counted at a distance of 0–500μm to Coll4^+^CNV of the indicated strains (*n* = 9–10 eyes; * *p* = 0.0002; ^†^
*p* = 0.0114; Mann–Whitney versus the normoxic C57BL6/J mice, hypoxic Thbs1^−/−^ mice were not significantly different from hypoxic C57BL6/J mice). (B) Quantification of the Coll4^+^CNV surface at d10 of normoxic and hypoxia-exposed (d4–d10) laser-injured mice of the indicated strains (*n* = 9–10 eyes; * *p* < 0.0001; ^†^
*p* = 0.0114; Mann–Whitney versus the normoxic C57BL6/J mice). (C) Representative images of Coll4 (red) and IBA-1 (green) immuno- stained RPE/choroid flatmounts of 10 day post laser-injured, 2 month-old, wildtype mice exposed to 10% O_2_ hypoxia (d4–d10) that were intravitreally injected at d4 and d7 with 2µL of PBS, or 2µL of PBS containing recombinant Thrombospondin 1 protein (TSP-1, 10µg/mL) and the quantification of subretinal IBA-1^+^MPs per impact on the RPE counted at a distance of 0–500μm to Coll4^+^CNV at d10 (*n* = 9–10 eyes; * *p* = 0.0001; versus the hypoxic PBS-treated mice). (D) CNV surface at d10 of the treated, hypoxia-exposed (d4–d10), laser-injured mice (*n* = 9–10 eyes; * *p* = 0.00143; Mann–Whitney versus the hypoxic PBS-treated mice). All values are reported as mean  ±  SEM. PBS: Phosphate buffered saline; rTSP-1: Recombinant thrombospondin 1; scale bar = 400 μm.

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
