# Peer review of "Hypoxia Inhibits Subretinal Inflammation Resolution Thrombospondin-1 Dependently"

_ijms, 2022, doi:10.3390/ijms23020681_

Round 1
Reviewer 1 Report
The authors show that TSP1 gene is downregulated in monocytes of mice after hypoxia, and that TSP1 might play an important role in resolution of inflammation. My questions and suggestions are outlined below:
- The authors use OTX2 as a marker for RPE cells, but OTX2 is also expressed in other cells of the retina like photoreceptors and bipolars. Can the authors use other markers to confirm that the RPE cultures are indeed pure, maybe by staining or MITF, ZO-1 or RPE65?
- The authors say they have inflammation in the retina after injury, but no images are shown. The authors also state multiple times that inflammation was seen next to the RPE layer but not next to endothelial cells, but no images were provided as proofs. Including a 10x image showing the injury site would help to understand.
- Can the authors also provide low power images of the staining in A to better depict how the phenotype was distributed?
- How many mice were used for the quantification? Please plot animal numbers on the graph sand provide representative images of the areas used for quantification
- Figure 1- C and D, when was this quantification done? And was the effect of hypoxia transient? For example, the IBA1 phenotype seen in A, was it also seen 20 days or 2 months later?
- Figure 2- what is the difference between B and C? (not clear from figure labels, please see #10)
- Figure 2A- why was analysis done 40 hours versus figure 1, where hypoxia was induced for longer?
- How does the effect in 3C compared to uninjured retinas? Does it fully rescue the phenotype?
- How was the dosage of recombinant TSP1 optimized? How long after injection was the quantification done? did the TSP1 actually rescue the phenotype for long periods of time or was it a temporary relief?
- The figure legends for figures 2 and 3 need labels, it is unclear which sentences refer to which panels in the figures
- Not that crucial but- The abstract, introduction and discussion are heavily dedicated to AMD, but a direct correlation between AMD and hypoxia is not shown in this manuscript. In fact, there is no data on AMD in this manuscript. While the inflammation in this study may recapitulate AMD pathogenesis, and it is good to emphasize the applicability of their study, extended explanation and discussion of AMD is slightly tangential. More background on inflammation, and the immune cells described in this study and how inflammation is resolved in healthy versus diseased retinas would be helpful.
Author Response
Reviewer 1
The authors show that TSP1 gene is downregulated in monocytes of mice after hypoxia, and that TSP1 might play an important role in resolution of inflammation. My questions and suggestions are outlined below:
We would like to thank the reviewer for taking the time to review our manuscript!
- The authors use OTX2 as a marker for RPE cells, but OTX2 is also expressed in other cells of the retina like photoreceptors and bipolars. Can the authors use other markers to confirm that the RPE cultures are indeed pure, maybe by staining or MITF, ZO-1 or RPE65?
We did not try to claim that OTX2 is specific for RPE cells. We use it in the co-culture system as it allows to differentiate PU1+OTX2neg Mo and OTX2+PU1negRPE cells. Both of which are nuclear markers and make the automated quantification easier. In these co-culture experiments we use primary RPE cells that we characterized in previous publications (Mathis et al., 2017; Touhami et al., 2018). Indeed, we showed that these primary cell cultures form tight junctions, they phagocytose photoreceptor outer segments, stain positive for ZO1, and express retinol dehydrogenase 5, transthyretin, transferrin. Please find above additionally an OTX2/RPE65 double-labelling.
To make clear to the reader that these marker were used in the particular co-culture model context, we changed the sentence in the results section to
“….that allows the differentiation of PU1+OTX2neg Mo and OTX2+PU1negRPE cells in this cell culture system.
And added to the M&M the following sentence:
We previously characterized these primary RPE cell cultures, showing that they form tight junctions and phagocytose photoreceptor outer segments, stain positive for ZO1, and express retinol dehydrogenase 5, transthyretin, transferrin (Mathis et al., 2017; Touhami et al., 2018)
- The authors say they have inflammation in the retina after injury, but no images are shown. The authors also state multiple times that inflammation was seen next to the RPE layer but not next to endothelial cells, but no images were provided as proofs. Including a 10x image showing the injury site would help to understand.
- Can the authors also provide low power images of the staining in A to better depict how the phenotype was distributed?
Physiologically the subretinal space is devoid of IBA1+MPs and of Coll4+ vessels. After the laser-injury IBA1+MPs infiltrate the subretinal space, which we call here inflammation (infiltration and numerical increase of MPs). It is depicted in all the IBA/Coll4 stained flatmount pictures in Figure 1 and Figure 3. In the model, we distinguish the RPE-denuded lesion that contains the CNV growth, and its RPE containing surroundings. In terms of MP density, hypoxia only influences the MP density on the immunosuppressive RPE, not in the RPE denuded lesion itself. In terms of how best to represent this, we chose a magnification that allows the visualization of the individual IBA1+MPs, depicts the Coll4+CNV lesion and shows some of the surrounding area. To make the understanding of the results easier we rewrote the description of the result section:
“The subretinal space can be visualized by flat-mount preparations and is physiologically avascular and devoid of IBA1+MPs. Laser-injury induces the infiltration of subretinal IBA1+MPs, with a maximal recruitment three to four days after the injury, followed by an inflammation resolution phase characterized by dwindling MP numbers and choroidal neovascularization (CNV) formation (Lavalette et al., 2011). 50-70% of the lesional MPs are derived from circulating Mos and the remainder from resident macrophages (microglial cells and choroidal macrophages) (supplementary Fig.5 of Beguier et al. (Beguier et al., 2020)) (Caicedo et al., 2005). Depletion of circulating Mo (Beguier et al., 2020; Caicedo et al., 2005; Sakurai et al., 2003a ) and inhibition of Mo-recruitment (Liu et al., 2013; Luhmann et al., 2009; Robbie et al., 2016 ; Sakurai et al., 2003b; Tsutsumi et al., 2003) very significantly inhibits CNV formation. Importantly, the infiltrating MPs are observed within the injured tissue in close contact with the forming CNV, but also in the subretinal space, adjacent to physiologically immunosuppressive RPE cells, that are potent inducers of MP death (Calippe et al., 2017; Lavalette et al., 2011 ; Levy et al., 2015a ; Mathis et al., 2017).
To evaluate the influence of hypoxia on subretinal neuro-inflammation we exposed laser-injured mice to 10% or 20.9% of ambient oxygen. As young, otherwise healthy, mice quickly adapt to hypoxia, increasing their hematocrit (Yu et al., 1999) we either exposed the mice during the early phase (d1-4) or during the inflammation resolution phase, when MP elimination exceeds the recruitment rate (d4-10; experimental design Fig. 1A). Quantification on RPE/choroidal flatmounts of subretinal IBA-1+ (green) MPs on Collagen-4+ (Coll4+, red) CNV and on the surrounding RPE (0–500 μm from the lesion) revealed that hypoxia did not significantly alter the density of subretinal MPs in the RPE-denuded lesions, directly adjacent to the endothelial cells of the CNV. However, it significantly increased the accumulation of IBA1+MPs that accumulate on the immunosuppressive RPE at d4 (190% increase) and induced a continued accumulation of MPs from d4 to d10, when the MP accumulation started to resolve in normoxic mice (460% increase at day 10; Fig. 1B-D). Additionally, the size of Coll4+CNV was significantly increased at d10 (Fig. 1E).
- How many mice were used for the quantification? Please plot animal numbers on the graph sand provide representative images of the areas used for quantification
The number of eyes used in each graph are indicated in the figure legends. Representative images are provided for flatmounts of normoxic and hypoxic 10 day post laser injury mice.
- Figure 1- C and D, when was this quantification done? And was the effect of hypoxia transient? For example, the IBA1 phenotype seen in A, was it also seen 20 days or 2 months later?
The x axis (d4, d10) as well as the figure legend indicates when the quantification of 1C and 1D (now 1D and E) were performed. In the long term, the laser-induced lesions will further contract and slowly be covered by the RPE cells in normoxic conditions (PMCID: PMC2664845). In our experiments exposure of these young adult mice to longer periods of time had no additional effect as they adapt to the hypoxic environment (as mentioned in the text).
Figure 2- what is the difference between B and C? (not clear from figure labels, please see #10)
We are sorry for this mistake. It seems that the letters indicating the different panels were lost in the pdf layout. In the new manuscript this is corrected and cartoons were added to the figure to help the reader understand the experimental design.
- Figure 2A- why was analysis done 40 hours versus figure 1, where hypoxia was induced for longer?
An explanation for this timing was added to the results section of the corrected manuscript:
« A 40h exposure time was chosen for this experiment as normal mice quickly adapt to hypoxic conditions and to detect differences that occur prior to the inflammatory changes observed in figure 1. »
- How does the effect in 3C compared to uninjured retinas? Does it fully rescue the phenotype?
As mentioned above, uninjured retinas do not have IBA1+ MPs or CNV in their subretinal space. TSP1 injections (depicted in 3C) fully rescue the exaggerated additional inflammation induced by the hypoxia as clear from the graphs.
- How was the dosage of recombinant TSP1 optimized? How long after injection was the quantification done? did the TSP1 actually rescue the phenotype for long periods of time or was it a temporary relief?
We have previously used recombinant TSP1 in other models and determined that an injection of 2µl of a 10µg/ml solution, which corresponds to a concentration of 80nM as TSP1 is a sizeable protein, induces an immune-suppressive effect. To better illustrate the experimental design we added cartoons to the new figure 3 (as for all the other figures). We have so far not evaluated later time points than d10. As mentioned above the inflammation and CNV of this model naturally diminishes over time.
- The figure legends for figures 2 and 3 need labels, it is unclear which sentences refer to which panels in the figures
We are sorry for this inconvenience. Somehow all our panel indications vanished in the figure legends in the layout that was sent to the reviewers. These errors were corrected
- Not that crucial but- The abstract, introduction and discussion are heavily dedicated to AMD, but a direct correlation between AMD and hypoxia is not shown in this manuscript. In fact, there is no data on AMD in this manuscript. While the inflammation in this study may recapitulate AMD pathogenesis, and it is good to emphasize the applicability of their study, extended explanation and discussion of AMD is slightly tangential. More background on inflammation, and the immune cells described in this study and how inflammation is resolved in healthy versus diseased retinas would be helpful.
We do agree with the reviewer that there is no human data in this manuscript. However, we used the model of laser-induced CNV that is characterized by inflammation and neovascularization similar to wet-AMD and widely used to model the disease. Additionally, as mentioned in the text, there are many reasons to suspect that local and systemic hypoxia plays a role in the pathogenesis of AMD as described in the discussion. These are the reasons why the relation of our results are discussed, but most of the discussion is directly treating inflammation resolution and hypoxia and discusses directly the results we obtained.

Reviewer 2 Report
Comments and Suggestions for Authors
Dear authors,
The article titled ‘Hypoxia inhibits subretinal inflammation resolution thrombospondin-1 dependently’ submitted by Touhami et al. to Molecular Sciences aim to demonstrate that the hypoxia-induced inhibition of mononuclear phagocytes elimination is dependent on downregulation of TSP-1.
MINOR AND MAJOR CORRECTIONS NEEDED:
ABSTRACT: Abstract is comprehensive and clear. Essential information relevant to the findings of the study is incorporated. However, in lines 19 and 22, Thbs-1 and TSP1 require to be defined, respectively. In addition, I suggest to revise the statement in lines (20-23).
INTRODUCTION: The introduction is well written however, a more extensive introduction about oxygen metabolism of the retina and relative impairments in AMD is required. In particular, as AMD is a multifactorial disease, it is important to state that hypoxia is one of the numerous conditions that retina might be affected during AMD. (as instance: https://www.ncbi.nlm.nih.gov/pmc/articles/PMC3950832/)
Furthermore, the state of art of TSP1 oxygen dependency and neovascularization need to be described, clarifying the step forward of your study.
(as instance: https://pubmed.ncbi.nlm.nih.gov/9851743/; https://iovs.arvojournals.org/article.aspx?articleid=2165117)
MATERIALS AND METHODS: This section is quite clear, however a scheme of the experimental protocols in the paragraph “Laser- injury model, intravitreal injections, and hypoxia” might help readers to understand the study.
RESULTS: This section is straightforward and quite clear. The authors have explained the section based on the results obtained and categorized them. However, the following needs attention.
- Figure 1 lacks “(A)” in line 209.
- Was assessed thinning of the photoreceptor layer within the impact? It might be important to assess whether the most oxygen-consuming cell of the retina, the photoreceptor, is affected by laser injury because oxygen availability might change (see for ref https://pubmed.ncbi.nlm.nih.gov/16212707/).
- Figure 1. “d1-d4” and “d4-d10” in the legend of B-D graphs are not consistent with “d4” and “d10” below the x-axes and it might be confusing. Please revise.
- Please update all captions with information about the error bars (standard error or standard deviation).
- In line 213 the hypoxic period is “d0-d4” while in the legend it is “d1-d4”. Please be consistent or clarify.
- Please revise the color code in figure 1, despite 2 different legends black bars can be confusing among graphs.
- Captions of figures 2 and 3 lack in reference letters, please revise.
- Figure 2B. In lines 295-296 both statistical tests are vs normoxic C57BL6/J mouse. However, is the hypoxic Thbs 1 -/- mouse statistically significant vs hypoxic C57BL6/J mouse?
- Caption 3. Please define “QuantificaTable 4.” in line 302.
DISCUSSION and CONCLUSION: The discussion is clear however, an extension of possible TSP1 mechanisms related to your results might add weightage to the article.
Author Response
Reviewer 2
Dear authors,
The article titled ‘Hypoxia inhibits subretinal inflammation resolution thrombospondin-1 dependently’ submitted by Touhami et al. to Molecular Sciences aim to demonstrate that the hypoxia-induced inhibition of mononuclear phagocytes elimination is dependent on downregulation of TSP-1.
We would like to thank the reviewer for taking the time to review our manuscript!
MINOR AND MAJOR CORRECTIONS NEEDED:
ABSTRACT: Abstract is comprehensive and clear. Essential information relevant to the findings of the study is incorporated. However, in lines 19 and 22, Thbs-1 and TSP1 require to be defined, respectively. In addition, I suggest to revise the statement in lines (20-23).
We correct this in the revised manuscript
INTRODUCTION: The introduction is well written however, a more extensive introduction about oxygen metabolism of the retina and relative impairments in AMD is required. In particular, as AMD is a multifactorial disease, it is important to state that hypoxia is one of the numerous conditions that retina might be affected during AMD. (as instance: https://www.ncbi.nlm.nih.gov/pmc/articles/PMC3950832/)
We added the following sentence and reference to the introduction:
« Hypoxia, among other contributing mechanisms such as oxydative stress and dysfunctional autophagy, has long been supposed to be an important trigger of AMD (Blasiak et al., 2014). »
Furthermore, the state of art of TSP1 oxygen dependency and neovascularization need to be described, clarifying the step forward of your study.
(as instance: https://pubmed.ncbi.nlm.nih.gov/9851743/; https://iovs.arvojournals.org/article.aspx?articleid=2165117)
The following sentence was added to the introduction: “Accordingly, TSP-1 deletion leads to increased subretinal inflammation accompanied by increased CNV in laser-injured mice (Beguier et al., 2020; Ng et al., 2009; Wang et al., 2012).”
These points are also more extensively addressed in the discussion, including a paragraph on the possible mechanism of Thbs-1 repression by hypoxia in monocytes:
“Hypoxia can activate macrophages via the transcription factors hypoxia-inducible factor 1α and 2α (HIF-1α, HIF-2α), NF-κB and AP-1 that is formed by c-Jun and c-Fos (Lewis et al., 1999 ; Murdoch et al., 2005 ; Rahat et al., 2011). Intriguingly, the regimen also led to a significant decrease in Thbs1- transcripts in Mo (Fig. 2). These alterations might be due to c-jun, which is increased by hypoxia (Ausserer et al., 1994) and represses Thbs1-tanscription (Mettouchi et al., 1994) »
MATERIALS AND METHODS: This section is quite clear, however a scheme of the experimental protocols in the paragraph “Laser- injury model, intravitreal injections, and hypoxia” might help readers to understand the study.
Cartoons summarizing the experimental design were added to each figure to facilitate the reading of the article.
RESULTS: This section is straightforward and quite clear. The authors have explained the section based on the results obtained and categorized them. However, the following needs attention.
- Figure 1 lacks “(A)” in line 209.
Indeed, many of the alphabetical panel labeling went missing in the layout that was sent out for review. These errors were all corrected in the revised manuscript
- Was assessed thinning of the photoreceptor layer within the impact? It might be important to assess whether the most oxygen-consuming cell of the retina, the photoreceptor, is affected by laser injury because oxygen availability might change (see for ref https://pubmed.ncbi.nlm.nih.gov/16212707/).
We did not evaluate the degeneration of the photoreceptors surrounding the laser impacts, which might be an interesting subject for future studies.
Figure 1. “d1-d4” and “d4-d10” in the legend of B-D graphs are not consistent with “d4” and “d10” below the x-axes and it might be confusing. Please revise.
Thank you these errors were corrected
- Please update all captions with information about the error bars (standard error or standard deviation).
We added this information to the legends. It was already mentioned in the M&M.
- In line 213 the hypoxic period is “d0-d4” while in the legend it is “d1-d4”. Please be consistent or clarify.
- Please revise the color code in figure 1, despite 2 different legends black bars can be confusing among graphs.
- Captions of figures 2 and 3 lack in reference letters, please revise.
These shortfalls were all corrected in the revised manuscript
- Figure 2B. In lines 295-296 both statistical tests are vs normoxic C57BL6/J mouse. However, is the hypoxic Thbs 1 -/- mouse statistically significant vs hypoxic C57BL6/J mouse?
Hypoxic Thbs1-/- mice were not significantly different from hypoxic C57BL6/J mice. We added this information to the figure legend
- Caption 3. Please define “QuantificaTable 4.” in line 302.
“QuantificaTable 4.” Was removed from the legend, it must have appeared here accidentially.
DISCUSSION and CONCLUSION: The discussion is clear however, an extension of possible TSP1 mechanisms related to your results might add weightage to the article.
We added a sentence in that sense to the discussion:
« …the deregulation of TSP-1 expression and ensuing increased subretinal inflammation and production of pathogenic cytokines. »
Submission Date
30 September 2021
Date of this review
12 Nov 2021 11:28:59
Bottom of Form
© 1996-2021 MDPI (Basel, Switzerland) unless otherwise stated
Disclaimer Terms and Conditions Privacy Policy

Round 2
Reviewer 2 Report
The study is better described and sufficiently straightforward.
Author Response
We would like to thank the reviewer for his time and help to improve the manuscript.